# Protoplast-Based Regeneration Enables CRISPR/Cas9 Application in Two Temperate *Japonica* Rice Cultivars

**DOI:** 10.3390/plants14132059

**Published:** 2025-07-05

**Authors:** Marion Barrera, Blanca Olmedo, Matías Narváez, Felipe Moenne-Locoz, Anett Rubio, Catalina Pérez, Karla Cordero-Lara, Humberto Prieto

**Affiliations:** 1Biotechnology Laboratory, La Platina Research Station, National Institute of Agriculture (INIA), Santiago 8831314, Chile; marion.barrerac@utem.cl (M.B.); bolmedo@inia.cl (B.O.); anettrubio@ug.uchile.cl (A.R.); 2Natural Sciences, Mathematics, and Environment Faculty, Metropolitan Technological University, Santiago 8330526, Chile; mnarvaezs@utem.cl (M.N.);; 3Sciences Faculty, Universidad Santo Tomás, Santiago 8370003, Chile; f.moennelocoz@alumnos.santotomas.cl; 4Rice Breeding Program, Quilamapu Research Station, National Institute of Agriculture (INIA), Av. Vicente Méndez 515, Chillán 3780000, Chile; kcordero@inia.cl

**Keywords:** cold-adapted rice, Chilean rice, new genomic techniques, transfection, feeder extract, alginate encapsulation

## Abstract

Rice (*Oryza sativa* L.), a staple food for over half of the global population, plays a pivotal role in food security. Among its two primary groups, *japonica* and *indica*, temperate *japonica* varieties are particularly valued for their high-quality grain and culinary uses. Although some of these varieties are adapted to cooler climates, they often suffer from reduced productivity or increased disease susceptibility when cultivated in warmer productive environments. These limitations underscore the need for breeding programs to incorporate biotechnological tools that can enhance the adaptability and resilience of the plants. However, New Genomic Techniques (NGTs), including CRISPR-Cas9, require robust in vitro systems, which are still underdeveloped for temperate *japonica* genotypes. In this study, we developed a reproducible and adaptable protocol for protoplast isolation and regeneration from the temperate *japonica* cultivars ‘Ónix’ and ‘Platino’ using somatic embryos as the starting tissue. Protoplasts were isolated via enzymatic digestion (1.5% Cellulase Onozuka R-10 and 0.75% Macerozyme R-10) in 0.6 M AA medium over 18–20 h at 28 °C. Regeneration was achieved through encapsulation in alginate beads and coculture with feeder extracts in 2N6 medium, leading to embryogenic callus formation within 35 days. Seedlings were regenerated in N6R and N6F media and acclimatized under greenhouse conditions within three months. The isolated protoplast quality displayed viability rates of 70–99% within 48 h and supported transient PEG-mediated transfection with GFP. Additionally, the transient expression of a gene editing CRISPR-Cas9 construct targeting the *DROUGHT AND SALT TOLERANCE* (*OsDST*) gene confirmed genome editing capability. This protocol offers a scalable and genotype-adaptable system for protoplast-based regeneration and gene editing in temperate *japonica* rice, supporting the application of NGTs in the breeding of cold-adapted cultivars.

## 1. Introduction

Rice (*Oryza sativa* L.) is a staple food for more than half of the world’s population, playing a crucial role in global food security. The species is broadly classified into two main groups, *japonica* and *indica*, which differ in their growing conditions and grain characteristics. Temperate *japonica* rice varieties are highly valued for their unique culinary applications and can be considered complimentary to the production of tropical subgroups in terms of their texture and high-quality grain [1]. However, despite being formerly classified as a group with small and highly specialized markets, there is an increasing demand for temperate *japonica* varieties. Today, these varieties are considered to have great commercial potential [2].

Although temperate *japonica* rice genotypes are widely produced in colder regions, where the lower temperatures can affect the plant’s reproductive phase, these materials have shown less adaptability and productivity and increased susceptibility to diseases when grown in tropical areas [1]. To enhance the performance and resilience of these varieties, breeding programs and biotechnological tools are essential to develop varieties with improved adaptability, productivity, and disease resistance [3]. However, most of these tools have not yet been enabled [4]. For instance, the successful incorporation of advanced techniques into plant breeding programs hinges on critical technical components, starting from adequate in vitro culture platforms that support access to New Genomic Techniques, including CRISPR-Cas9.

*Agrobacterium*-mediated gene transfer methods are crucial for initiating CRISPR-Cas9 gene editing [5]. The application of the same protocols that are used for genetic transformation, particularly those based on the use of multicellular explants, soon revealed certain limitations of gene editing that had not previously received significant attention [6]. Chimerism arises when only portions of the regenerated plant originate from a gene-edited cell, presenting difficulties in traditional tissue culture techniques that employ multicellular starting explants such as somatic embryos and organogenesis [7]. The use of multicellular explants as the initial material in experimental processes increases the likelihood that new shoots will develop from cells that are inaccessible to the gene transfer machinery rather than from a single cell that underwent the editing process, thus elevating the risk of unedited cell groups being recovered [8]. Rice was initially described as resistant to *Agrobacterium*-mediated gene transfer due to its lack of susceptibility to infection [9]. However, efficient protocols for this method have only been developed after considerable effort, representing a constant limitation and necessitating permanent construction based on previous procedures [10]. In this regard, methods aiming at the generation of uniformly edited individuals still depend on the continuous development of procedures that, by means of trial and error, will facilitate the induction of totipotency in different genotypes within a species [6].

Currently, protoplasts are considered a powerful technology for rapidly validating the mutagenesis efficiency of various RNA-guided endonucleases, promoters, sgRNA designs, or Cas proteins and can serve as a platform for DNA-free gene editing [11]. Protoplasts, which are plant cells devoid of cell walls, have a unique capacity to take up various macromolecules such as DNA, RNA, and proteins effectively. After this process, the regenerated plants will, in most cases, derive from a single cell, resulting in increased genetic uniformity. This approach meets additional biosafety standards that require DNA-free gene editing, a technique that is used to avoid the regulatory and public concerns associated with modifying an organism’s genetic material without introducing foreign DNA [12]. In this way, protoplast techniques can be considered to enable CRISPR/Cas9 systems that use ribonucleoprotein complexes instead of DNA vectors. Additionally, the use of transient transformation of protoplasts can circumvent transgenesis (i.e., the integration of genetic material from one organism into the genome of another), such as advanced delivery techniques using modified T-DNAs that allow for the intensive transient expression of CRISPR-Cas9 editor reagents [13].

The relevance of developing reliable methods for regenerating rice plants from protoplasts in temperate *japonica* rice varieties was formerly described, using somatic embryos of the Taipei 309 [14] and calli from Raldon and Baldo [15] as starting materials. Later, comprehensive protocols and advancements in the technique for this subgroup included Nipponbare [16]. These works showed the key factors that influence the feasibility of producing rice protoplast systems and included the starting material (type and age), enzyme treatment (choice, concentration, time of digestion), osmotic conditions (avoiding bursting or shrinking), purification procedures (debris elimination), and viability and regeneration efficiencies (i.e., culture conditions) [17]. Consequently, the development of a complete procedure leading to reproducible regeneration of temperate *japonica* rice plants stills requires careful optimization and handling [18], and although optimized methods and varieties have shown promising results, they often have significant limitations when applied to less studied ecotypes, such as those from marginal regions or those with specific agronomic characteristics [19]. These discrepancies highlight the need to adapt or develop new protocols that enable us to overcome genotypic compatibility barriers and optimize the regeneration of complete plants from protoplasts in a wider range of rice subtypes, including temperate *japonica* genotypes.

With the aim of establishing foundational biotechnological tools to support the Chilean Rice Breeding Program, entirely based on the *japonica* germplasm, we previously developed effective methodologies for obtaining and regenerating embryogenic callus from seeds of elite local temperate *japonica* varieties (‘Platino’, ‘Cuarzo’, ‘Esmeralda’, and ‘Zafiro’), using 2N6 medium supplemented with 2,4-dichlorophenoxyacetic acid (2,4-D) [20]. From that platform, we employed these somatic embryogenesis systems as a starting point to develop protoplast-based technologies tailored to this class of cultivars.

The rice cultivars ‘Platino’ and ‘Onix’, which were used in this study, represent key advances in Chilean temperate *japonica* breeding. ‘Platino’ is a semi-late-maturing, medium-grain variety with reduced plant height, high yield potential, and improved grain quality compared with older cultivars [2]. Its translucent grains, low chalkiness, and desirable cooking texture make it particularly suited for processed food products including sushi, risotto, and rice pudding [21]. ‘Onix’ is Chile’s first black rice cultivar, developed for its medium-sized purple grains that are rich in anthocyanins, with promising applications in functional foods and niche markets [22]. These cultivars were selected for their agronomic stability and favorable responsiveness to in vitro culture [20], which are critical for the development of genotype-specific regeneration protocols aimed at precision breeding.

In the present study, we expanded the establishment of the conditions for the isolation and regeneration of protoplasts from embryogenic callus cultures for ‘Platino’ and ‘Ónix’. The inclusion of gene transfection and editing into these procedures demonstrated their suitability as facilitators of genetic improvement for these varieties, thereby contributing to the development of new genotypes, which is useful for breeding and expanding the potential for adding new genetics into this subgroup.

## 2. Results

### 2.1. Protoplast Isolation

Competent protoplasts that were capable of regenerating in ‘Platino’ and ‘Ónix’ varieties were developed from young embryonic cultures, generated as previously described for the temperate rice genotypes ‘Platino’, ‘Cuarzo’, ‘Esmeralda’, and ‘Zafiro’ [20]. These embryogenic cultures show a tendency to double their mass within 30 d and continue to proliferate for 6 months, with subcultures occurring every 2 weeks in 2N6 medium. Under this regime, based on a typical use of 40 mature seeds as the initial material for somatic embryogenesis induction, approximately 5 g of embryogenic callus is obtained in both varieties. It was found that careful selection of the optimal callus material for protoplast isolation was essential for obtaining intact cells. Calli with a friable consistency and a pale yellow color (Figure 1A), indicating high cellular density and embryogenic activity, were selected.

The protoplasts were isolated using an enzymatic digestion protocol that included the use of an enzymatic cocktail comprising 1.5% (*w*/*v*) Onozuka R-10 Cellulase and 0.75% (*w*/*v*) Macerozyme R-10, for which different masses and ages were evaluated. According to our evaluations, the optimal conditions for the isolation of abundant, spherical, and intact protoplasts were established through incubation for 18–20 h in the dark at 28 °C and 40 rpm. A visual indication of successful enzymatic digestion was a milky appearance of the solution after 18 h of gentle shaking (Figure 1B). Additionally, our findings showed that using 500 mg of embryogenic mass, which was cultivated and propagated in 2N6 medium for two months under long-day conditions (16 h light/8 h dark), enhanced the yield of intact protoplasts after enzymatic digestion.

Supplementary trials were conducted, in which younger calli (<4 weeks), smaller embryogenic masses (<500 mg), or shorter incubation periods with the enzymatic solution (<7 h) were evaluated, leading to a reduction in protoplast density, as suggested by the transparent enzymatic appearance of the mixture after treatment. In the latter case, the resulting protoplasts were primarily small and fragile and had a low division potential. Meanwhile, when using larger embryogenic masses (>500 mg) or prolonged digestions (>20 h), the enzymatic solution turned brown, suggesting an excess of undigested tissue and excessive breakdown of the cell wall, respectively.

After overnight incubation of the callus tissue in the enzymatic mixture for 18–20 h, the protoplasts were isolated from the residual tissue by means of filtration (Figure 1C) and centrifugation (Figure 1D) with an FW washing solution. This process resulted in the successful release of spherical protoplasts ranging from 5 to 15 μm in size (Figure 1E,F). The average yield of protoplasts derived from rice calli was 2.7 × 10^6^ ± 4.5 × 10^4^ cells per gram of fresh weight (g^−^^1fw^) for the Ónix variety and 3.8 × 10^6^ ± 6.6 × 10^4^ cells·g^−1fw^ for the Platino variety.

The viability of the rice protoplasts was assessed using fluorescein diacetate (FDA) staining (Figure 2, upper panel); FDA enters live cells and is enzymatically converted into fluorescein, making it a reliable method for identifying metabolically active cells [23]. Our viability assays showed that approximately 99% of the protoplasts maintained their viability 1 h post-isolation, with no apparent decrease at 24 h post-isolation. An estimated 30% decrease in viability was regularly observed 48 h post-isolation (Figure 2, middle panel).

### 2.2. Gene Transfer and Editing Applications in Isolated Protoplasts

Protoplasts are currently recognized as excellent platforms for gene transfer techniques, enabling the introduction of foreign DNA for functional studies, genetic improvement, and advancements in plant biotechnology. Regarding the future utility of these procedures, we established the technical compatibility of the obtained protoplasts for transient transfection procedures mediated by PEG-Ca^2+^. To evaluate the transfection competence, we monitored the expression of the reporter gene, *Green Fluorescent Protein* (*GFP*), which was included in the vector (pFLC-U), a modified T-DNA vector that can generate geminivirus DNA replicons in the cells [13]. At 24 h post-transfection, the GFP fluorescence results demonstrated an initial efficiency of approximately 77% in both the Platino and Ónix varieties (Figure 2, lower panel). By 72 h post-transfection, the efficiency decreased to 47% in ‘Platino’ and 33% in ‘Ónix’, accompanied by reductions in fluorescence levels of 70% and 60%, respectively, compared with our initial observations. These results highlight the potential of these protoplasts for transient gene expression studies while identifying time-dependent limitations in fluorescence retention.

The potential of employing gene editing in these cell systems was subsequently assessed through transient editing of the *DROUGHT AND SALT TOLERANCE* (*OsDST*) gene. This gene is known to play a critical role in plants’ responses to abiotic stresses, including drought and salinity, by regulating the stomatal density and water retention. A tailored gene editing construct, based on the sequence of Ónix, was developed to target the region encoding the EAR-C terminal of the gene product. After 24 h of transfection, Ónix protoplasts were used for DNA isolation, and the status of the *OsDST* gene was screened via PCR. While the wild-type protoplasts showed an amplification fragment of 555 bps, the transfected protoplasts displayed the expected double-cut edited band of approximately 261 bps (Figure 2C), with sequencing confirming the intended edit (Figure 2D). While these findings demonstrate the versality of the produced protoplasts as systems for gene transfer and editing applications, the next step involved analyzing the establishment of the complete regenerative process.

### 2.3. Protoplast Regeneration

To achieve regeneration, protoplasts were immobilized in calcium alginate beads; immobilization provides a supportive environment for protoplasts, protecting them from physical stress and dehydration while promoting their development into callus colonies. To increase regeneration, we included a feeder extract in the system, which proved to be an effective method for promoting the active and healthy development of encapsulated protoplasts. To prevent cellular dehydration, an encapsulation solution with an initial concentration of 0.6 M mannitol was used, which was gradually reduced to 0.1 M, maintaining the appropriate osmotic balance. This gradual reduction in mannitol is critical for maintaining osmotic balance, ensuring cellular stability, and preventing stress during the regeneration process.

Under the adjusted conditions, the first callus colonies became visible as early as 5 d after encapsulation (Figure 3), when the beads changed from a shiny to an opaque appearance. This change indicated that the protoplasts were active and beginning to form new cellular structures. Approximately 15 d post-encapsulation, a significant formation of microcallus colonies was observed inside the alginate beads. These microcalli continued their development and, between days 20 and 25 post-encapsulation, reached a size that was large enough to break the surface of the pearls (Figure 3). They continued to proliferate for an additional 10 d in AA medium supplemented with 0.1 M mannitol, resulting in the formation of primary embryogenic calli. Following this period, they were transferred to a solidified 2N6 (2,4-D) medium to initiate the regeneration steps.

We observed that coculturing with a feeder cell solution played a critical role in the success of these assays. The absence of this solution in the liquid culture with an AA medium of protoplasts encapsulated in calcium alginate beads resulted in a lack of callus colony formation, as evidenced by an evaluation at 30 d, and no changes were observed during the four months of monitoring (Appendix A).

### 2.4. Plant Regeneration

After 10 d of transferring the embryogenic calli to the 2N6 culture medium, the microscopic analysis of the cultivated explants showed adequate multiplication and propagation in the two examined genotypes (Figure 4). When these embryonic masses were transferred to the germination medium N6R, the first germination areas were generated after 15 d, while the green shoot-like structures became visible between 14 and 20 d after germination. The active shoots were isolated and cultured in the elongation medium N6F to promote growth, leading to the development of the first seedlings. Ten days after transferring the seedlings to the N6F medium, they were transferred to an ex vitro acclimatization room.

The implemented protocol led to the production of 12 seedlings per gram of starting somatic embryogenic callus in the Platino variety. In contrast, ‘Ónix’ showed a slightly higher yield, producing 20 seedlings per gram of tissue. These plants were acclimatized under chamber conditions for six weeks and finally moved to a greenhouse (Appendix A). The process, from the isolation of protoplasts to the obtainment of seedlings that were ready to be acclimatized ex vitro required 2–3 months. The protocol yielded an average of 12 and 20 regenerated seedlings per gram of embryogenic callus for ‘Platino’ and ‘Ónix’, respectively. Although the number of viable callus colonies per gram was not determined, these outputs reflect consistent regeneration performance and suggest cultivar-specific differences in the responsiveness to the protocol.

## 3. Discussion

The effectiveness of protoplast isolation and plant regeneration procedures depend on several factors, including the genotype, donor tissue source, and culture system [6,24,25]. In this work, we successfully developed a procedure to regenerate temperate *japonica* individuals from protoplasts derived from embryogenic calli in the elite varieties Platino and Ónix, cultivated in the southernmost productive area in the world. The procedure is based on the use of nurse cells for the recovery and generation of protoplast-derived calli, combined with a “bead-type” culture.

*Japonica* rice protoplasts that are suitable for the subsequent culture of protoplast-derived calli and plantlet regeneration were first described in ‘Nihonbare’ and ‘Sasanishiki’ [26]. Later, somatic embryo cells from ‘Taipei 309’ proved useful for producing protoplasts with regenerative potential [26,27]. Different sources, such as calli or cultured cells, have been successfully evaluated for *japonica* rice [28]. For ‘Platino’ and ‘Ónix’, the source of explants for enzymatic digestion was established from somatic embryo cultures from seed materials [20]. These seed-derived calli were generated by means of subcultivation on an induction medium (2N6). This choice facilitates the handling and disinfection of the starting material (seeds), reduces the risk of contamination, and ensures continuous supply for protoplast isolation. While these earlier studies established the foundational use of cultivars such as ‘Nihonbare’, ‘Sasanishiki’, and ‘Taipei 309’ for protoplast regeneration, they often relied on specific genotypes with high embryogenic potential or required complex purification steps such as density gradients. In contrast, the present study demonstrates a simplified and reproducible protocol for protoplast isolation and regeneration in two Chilean-adapted *japonica* cultivars, ‘Platino’ and ‘Ónix’, using seed-derived somatic embryos maintained in a 2N6 medium. Notably, the protocol does not require genotype-specific feeder lines or sucrose gradients, yet still supports plantlet regeneration, underscoring its adaptability to regionally important germplasms.

The generation of protoplasts in previous works used enzyme mixtures involving cellulase and macerozymes as the main components of the cell wall degradation process. Fujimura et al. [26] used a higher concentration of cellulase (4%) and macerozymes (1%) than Yamada et al. [27] and Masuda et al. [28] (who used 2% and 0.5%, respectively). Slight differences in cellulase proteins can also be observed for these works, which use Cellulase Onozuka R-10 or RS; Cellulase Onozuka RS has higher enzyme activity, a slightly different optimal pH and temperature range, improved solubility, and additional enzyme activities compared with Cellulase Onozuka R-10 [29]. When developing the procedure for ‘Platino’ and ‘Ónix’, we considered the use of cellulases with lower specific activity and reduced side activities such as xylanase, α-amylase, hemicellulase, pectinase, and protease [29]. In addition, the efficiency of protoplast isolation varies considerably among ecotypes and genotypes [6].

In our study, the optimal conditions for the enzymatic digestion of rice callus tissue in the evaluated varieties were empirically established, achieving effective degradation of the cell walls and maintaining the viability of the protoplasts within a period of 18 to 20 h. Shorter incubations (3–7 h) resulted in incomplete digestion and lower yields, while periods exceeding 20 h deteriorated the quality of the protoplasts. These findings contrast with other studies that report similar yields following less than five hours of digestion; these differences can be attributed to a distinct cell wall composition, the age of the starting material (such as tissues from stems and sheaths that are less than 12 d old) [19,30], or younger calli [31]. In general, young materials exhibit a lower content of rigid polysaccharides and a higher content of pectins, while embryogenic calli exhibit an increase in cellulose, lignin, and suberin, which gives them rigidity for embryonic development [32]. An example of this is the use of seedlings of up to 15 d of age, which produced a number of viable protoplasts compared with those obtained in our study, while more mature seedlings drastically reduce both the yield and quality of the protoplasts due to the strengthening of the cell walls [33].

Uniform protoplast preparation by means of the density gradient has been suggested to improve protoplast regeneration capability [28]. It is essential to note that, unlike most of the studies mentioned above on the efficient isolation of protoplasts, both in rice and other species, our approach included a protocol that is based on the isolation of protoplasts without the need for a sucrose gradient; the use of embryogenic masses kept in 2N6 medium [20] for two cycles of 15 days not only maintained the viability in the generated protoplasts but also yielded cells that were free of contamination with cellular debris or undigested cells, which simplified the process.

The culture conditions that are used for the proliferation of isolated protoplasts are fundamental to the success of their regeneration into plants. Technical improvements for culturing protoplasts have included their inclusion in agarose-solidified media [14,34] and the use of nurse cells, either included in an agar layer or combined with a “bead-type” culture [15,35]. These elements have been beneficial for colony formation and the subsequent regeneration of several *japonica* genotypes. One of the first reports on the use of feeders in the successful regeneration of *japonica* rice plants (“Radon” and “Baldo”) from protoplasts demonstrated cell growth using a Kao medium, but only when the appropriate feeder layer—based on the *indica* (“IR522” or “IR45”) cell extracts, prepared and maintained in LS medium—was present [15]. In our study, we used a combination of feeder cultures consisting of somatic embryo cells derived from “Cuarzo”, another *japonica* variety with low embryogenic potential [20], and the bead-type protection strategy of the protoplasts. Encapsulating protoplasts in alginate beads has been shown to be critical in several other systems [36], as the beads provide a barrier to separate protoplasts from feeder cells, helping avoid contamination from feeder cells during regeneration.

The inclusion of feeder extracts in protoplast culture has proven to be fundamental for promoting cell division and callus growth in banana protoplasts [37] and various subspecies of rice [35,38,39]. These findings are consistent with those from our study, where we discovered that cocultivation with suspension feeder cells of the same species as the cultivated protoplasts was essential to stimulate the efficient division of isolated protoplasts, allowing for the formation of callus colonies that eventually regenerate into embryos and, ultimately, into complete plants. This is attributed to the availability of growth factors, mainly proteins, in the protoplast culture, which are essential for signaling, cell division, and the development of microcalli. In our case, we selected a somatic embryogenic line from a less regenerative genotype (var. Cuarzo; [20]) as the feeder source. On the other hand, it has been identified that the source of nitrogen in the culture system is a critical factor for the sustained division of protoplasts [40] for the development of cell suspension cultures. Our results suggest that the use of AA medium, which provides a mixture of amino acids as the sole nitrogen source, facilitated an early and continuous release of protoplasts starting from the fifth day of cultivation. This led to the development of microcalli at a high frequency, which were able to germinate and regenerate complete plants.

Plant protoplasts are valuable tools in plant biotechnology, playing a fundamental role in gene characterization and plant modification through gene transfer or genome editing [41,42,43]. These applications contribute significantly to crop improvement. For the varieties assessed in this work, these procedures enable the use of these tools in genotypes that are receiving increasing attention not only in temperate–cold regions but also in new areas where their introduction has been complex [1]. We evaluated the quality of the obtained protoplasts, as well as the transfection efficiency for exogene expression and gene editing. In our preparations, almost 100% of the cells emitted a very bright FDA signal 24 h after isolation. This provides flexibility to the protocol, as it offers a time window after isolation to conduct experiments, such as transfections. These results complement previous research that reports similar findings based on other rice tissues [19,30].

Moreover, our preliminary evaluations showed a transfection rate of 77%, with a brief transient expression that was mediated by the used vector, which decreased at 72 h post-transfection. This was comparable to values reported by Zhang et al. [19]. This was predictable, given that the used vector is based on geminivirus, which implies that the introduced DNA does not integrate into the cellular genome, thus affecting the stability and persistence of the expression [44]. These findings demonstrate the feasibility of using the protoplasts obtained through the method described in our study for future transfection experiments. However, additional studies are required to optimize the transfection protocol for the evaluated rice varieties.

The described protocol showed limited genotype dependence and worked efficiently for both varieties—Onix and Platino—responding similarly in the processes of callus induction, isolation, and protoplast regeneration. Both varieties belong to the *japonica* subspecies, which tend to show more homogeneous behaviors compared with *indica* due to their lower genetic diversity [45]. These varieties are genetically related, resulting in a similar structure and chemical composition of the cell walls of the cultivars.

## 4. Materials and Methods

### 4.1. Plant Material and Tissue Culture Media

Seeds of the *japonica* rice (*Oryza sativa* L.) vars. Platino and Ónix were obtained from the Rice Breeding Program at INIA-Chile (Quilamapu Research Station) and used for somatic embryogenesis (SE) procedures. The compositions of the culture media used in this work are provided below (Table 1), and the preparation details for the solutions are listed in Appendix A.

### 4.2. Somatic Embryogenesis (SE) Callus Induction

Procedures to induce SE in both varieties were designed as previously described by Barrera et al. [20]. For zygotic embryo isolation, dehulled seeds were washed for 5 min with water and a neutral detergent and rinsed three times with water for 2 min. The washed seeds were surface-disinfected for 1 min using ethanol 76% (*v*/*v*), followed by three washes for 2 min in sterile water, transferred into a 20% (*v*/*v*) bleach plus two drops of Tween-20, and incubated for 20 min at 80 rpm in an orbital shaker. Afterward, the seeds were washed with sterile water six times for 2 min and dried using sterile filter paper (Cat. No. R1500; WSLabs, Quinta Normal, Santiago, Chile). The dried seeds were allowed to germinate via cultivation in 90 × 15 mm Petri dishes containing 2N6 medium for 3 d at 27 ± 1 °C and under a 16 h/8 h (light/dark) photoperiod. Embryos were rescued and cultured in the same medium for SE callus induction for 2 weeks. Explants were maintained in these conditions and subcultured every 15 d in the same medium up to somatic embryo formation. After two months, the embryogenic masses were used as source explants for protoplast isolation.

### 4.3. Protoplast Isolation from Embryogenic Callus Cultures

The enzymatic cocktail that was used for protoplast isolation consisted of 1.5% (*w*/*v*) Onozuka R-10 Cellulase (Cat. No. C8001; Duchefa Biochemic, Haarlem, The Netherlands) and 0.75% (*w*/*v*) Macerozyme R-10 (Cat. No. M8002; Duchefa Biochemic), dissolved in a Digestion Buffer containing 0.6 M mannitol, 10 mM KCl, 20 mM MES, and 10 mM CaCl_2_ at pH 5.7. The enzymatic solution was filtered through a 0.22 µm PES membrane syringe filter (Cat. No. 331011; Nest Biotech, Wuxi, China) prior to use. Five hundred milligrams of embryogenic calli, harvested 60 days after induction, were transferred to a sterile 90 × 15 mm Petri dish containing 5 mL of the filtered enzymatic cocktail and 2 mL of 0.6 M AA medium. The explants were incubated in this mixture for 18–20 h at 28 °C in darkness with continuous agitation at 40 rpm using an orbital shaker. After digestion, the resulting mixture was passed through a 40 μm cell strainer (Cat. No. 258366; Nest Biotech) into a sterile 15 mL Falcon tube and centrifuged at 500× g for 8 min at room temperature. The supernatant was discarded, and the protoplast pellet was gently washed twice with 800 μL of FW solution [0.6 M mannitol, 2 mM CaCl_2_, 5 mM MES, pH 5.7]. Finally, the purified protoplasts were resuspended in 100 μL of FW solution and pooled into sterile 1.5 mL Eppendorf tubes for subsequent analysis.

### 4.4. Determination of the Yield and Viability of Protoplasts

The protoplasts were resuspended in FW solution and counted using a Neubauer chamber under an Axio Lab A1 microscope (Zeiss, Oberkochen, Germany). The yield of protoplasts (Y) was expressed as the number of protoplasts g^−1fw^. The viability of the protoplasts was evaluated using FDA staining (Cat. No. F1303; Invitrogen, Waltham, MA, USA) after incubation for 5 min. The FDA solution in acetone was prepared by adding 0.01% FDA to the solvent, after which 100 µL of the protoplast suspension was stained with 1 µL of FDA and then observed under a microscope. The viability was calculated as the percentage of the number of protoplasts with green fluorescence/total number of protoplasts observed (*n* = 10). The data were calculated based on two independent biological replicates, with counts performed in at least three fields for each replicate.

### 4.5. Vectors

The vectors pFLC-U [20], for *Green Fluorescent Protein* gene expression, and pHUE411-DST, utilized for the *OsDST* gene editing, were used. For gene editing, the pHUE411 (Plasmid #62203, Addgene (Watertown, MA, USA) [46]) derivative was constructed by including gRNAs targeting the gene sequence [47] for Ónix. The sequence for this variety was obtained by means of PCR amplification of the EAR C- terminal region of the gene using the primers DST-Forward 5′-GATCGACATGCTCAACTGGAGG-3′ and DST-Reverse: 5′-AAGGAGTACGTACGCAGGCA-3′, which amplify a 342 pb fragment in the “Nipponbare” *japonica* reference genome. The amplified fragment was sequenced at Macrogen (Seoul, Republic of Korea), and its sequence was processed using the CRISPR P1.0 tool [48] to design the gRNA1 [GGATTAATCACACACGAGG] and gRNA2 [GCGGAGGTCAACTCGTACA], which were included in the gRNA cloning site of the pHUE411 vector by means of Golden Gate cloning [49].

### 4.6. PEG-Mediated Protoplast Transfections

Before transfection, protoplasts were pre-treated with MMG buffer [MES 4 mM, mannitol 0.4 M, MgCl_2_ 15 mM, pH 5.7], and plasmids were purified using the Zyppy Plasmid Miniprep kit (Cat. No. D4020; Zymo Research, Irvine, CA, USA). Then, 10 microliters of plasmid DNA (1 µg·µL^−1^) and 200 µL of the protoplast suspension were incubated in ice for 10 min. A fresh solution of polyethylene glycol (PEG-4000) [PEG 40% *p*/*v*, mannitol 0.6 M, Tris-HCl 2 mM, CaCl_2_ 5 mM pH 7.5] was added to the protoplast–DNA mix and incubated for an additional 20 min. The mixture was incubated for 10 min at room temperature in darkness, and then, this reaction was stopped by adding 1 mL of W5 buffer [NaCl 154 mM, CaCl_2_ 125 mM, KCl 5 mM, MES 2 mM, pH 5.7] and mixed by means of inversion. The protoplast suspension was centrifuged for 5 min at 200× *g* at room temperature to pellet cells. The supernatant from the centrifugation was discarded, and the protoplasts were resuspended in 200 µL of FW solution and incubated at room temperature in the dark for 3 d.

#### Transfection Evaluation

The expression of *GFP* in protoplasts that had been transfected with the pFLC-U vector was examined using a binocular microscope Axiolab 1 for LED epifluorescence (Zeiss), utilizing the fluorescent module that includes an excitation filter at 470 nm. Fluorescence photographs were taken with the Canon EOS Rebel T3 digital camera system with a GFP filter.

For *OsDST* double-cut editing, protoplast DNA isolation was carried out at 24 h post-transfection treatments using the Quick-DNA Miniprep Plus Kit (Zymo) according to the manufacturer’s instructions. The *OsDST* gene status was evaluated by means of PCR amplification using the primers EditDST-fwd (5’-CGCACCAGCACCATCTC-3’) and EditDST-rev (5’-AGGACTGATTGATCGATTACAAGG-3’), which amplify a 555 bp fragment of the wild-type *DST* gene and a 261 bp fragment when it is edited using double-cut editing. Control reactions were carried out using the *β-actin* gene (LOC4333919 actin-1-like [*Oryza sativa Japonica* Group (Japanese rice)]) as a parameter, as well as the primers Ac-Forward (5’-CTCAGGGTGGTTTCCGTTTA-3’) and Ac-Reverse (5’-GCAATGCCAGGGAACATAGT-3’), which amplify a fragment of 1192 bp.

### 4.7. Protoplast Ca^2+^ Alginate Embedding

Resuspended/processed cells were encapsulated using an alginate CaCl_2_ method. The wild-type protoplasts were encapsulated immediately after their isolation. Under laminar flow, the protoplast solution (typically 1 mL) was mixed with an equal volume of sterile 10% sodium alginate solution *m*/*v*. The protoplast/alginate mix was loaded into a sterile blue micropipette tip and expelled dropwise into a sterile 250 mL Erlenmeyer flask containing 30 mL of 50 mM CaCl_2_ solution. The formed beads were rinsed with 20 mL of FW solution. The alginate-embedded protoplasts were transferred into a new sterile 250 mL flask containing 40 mL of liquid 0.6 M AA medium and 10 mL of feeder extract. The suspension was incubated in the dark at 50 rpm and 25 ± 1 °C. After 14 d, 50 mL of AA medium without mannitol was added, reducing the initial concentration of mannitol to 0.3 M. The suspension was kept under culture as before, and after an additional 14 d, 50 mL of the suspension cultures were removed from the flasks and replaced by 50 mL of AA medium, reducing the mannitol concentration to a final 0.1 M. The embedded wild-type protoplasts were grown for 20 d, while the transfected protoplasts were cultured for 40 d.

### 4.8. Establishment of Conditioned Feeder (Nurse) Suspensions of Platino and Onix Rice Varieties

Approximately 1 g of embryogenic calli from both varieties were transferred to 250 mL Erlenmeyer flasks containing 20 mL of 2N6 liquid medium at 27 °C (in the dark) and 90–100 rpm. The concentration of mannitol in the culture medium was gradually increased for the osmotic conditioning of the feeder suspensions. Three biweekly subcultures were performed, where on each occasion, 10 mL of the suspension was removed and replaced with an equal volume of liquid medium 2N6, supplemented with 0.6 M mannitol. After the three subcultures, the solution from each flask was filtered using 40 μm filters (Cat. No. 258366, NEST Biotech, Wuxi, China) to avoid contamination with remnants of embryogenic callus and stored at −20 °C in 15 mL Falcon tubes.

### 4.9. Germination and Plant Regeneration

Once the protoplasts developed into callus colonies, usually 25 d after encapsulation, we manually transferred the microcalli to 90 × 15 mm Petri dishes containing 2N6 medium for callus propagation, culturing the dishes over 14 d at 27 ± 1 °C under a 16/8 h photoperiod. After 14 d, the embryogenic calli were transferred to the N6R regeneration medium for regular SE regeneration, as described by Barrera et al. [20]. Once the first shoots were obtained, they were transferred to the N6F rooting medium.

## 5. Conclusions

The method presented in this study enables the isolation of viable and morphologically intact protoplasts from somatic embryo-derived calli of two temperate *japonica* cultivars, ‘Platino’ and ‘Ónix’, with yields reaching over 10^6^ cells per gram and stability being maintained up to 48 h post-isolation. The approach avoids complex purification steps and relies on seed-derived calli and standard media, facilitating reproducibility and handling. The regeneration of fully acclimatized plants was achieved without visible phenotypic alterations, confirming the totipotent capacity of the isolated cells. Notably, both cultivars demonstrated consistent performance under identical culture conditions, with the differential plantlet yields suggesting a genotype-dependent response. Together, these findings reinforce the feasibility of applying this system to underutilized germplasms and establish a valuable platform for future applications, such as transient expression studies and gene editing in Chilean-adapted *japonica* rice.

## Figures and Tables

**Figure 1 plants-14-02059-f001:**
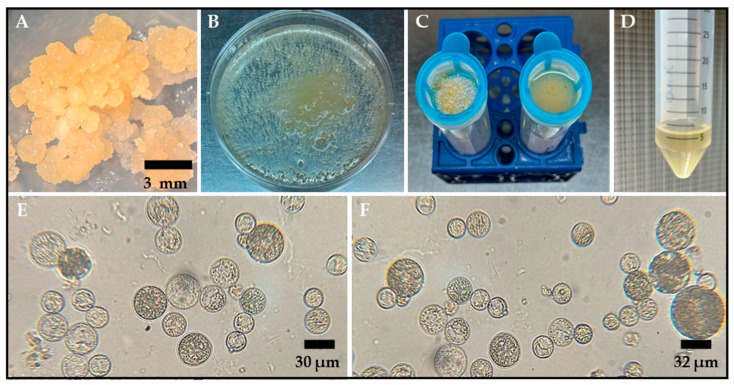
Protoplast isolation from embryogenic calli of the rice varieties Ónix and Platino. (**A**) Two-month-old embryogenic callus used as the starting tissue for protoplast isolation. (**B**) Embryogenic callus treated with an enzymatic solution to degrade the tissue’s cell walls. (**C**) Filtration through a cell strainer of the protoplast/enzymatic solution mixture after 18 h of digestion to separate large debris and damaged tissue. (**D**) The result of the centrifugation with FW washing solution of the material obtained from the previous filtration, where the obtained protoplasts are concentrated in the pellet. Bright-field microscope observations for protoplasts isolated 18–20 h after incubation in an enzymatic solution are shown for ‘Ónix’ (**E**) and ‘Platino’ (**F**). The bars indicate the approximate sizes specified.

**Figure 2 plants-14-02059-f002:**
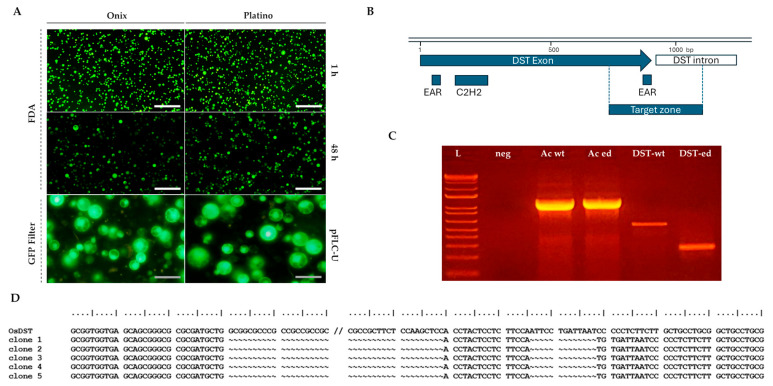
Evaluation of the viability and transfection competence using the pFLC-U vector and molecular confirmation of the editing of the *DROUGHT AND SALT TOLERANCE (OsDST)* gene in protoplasts isolated from rice embryogenic calli. (**A**) Protoplasts stained with 0.01% fluorescein diacetate (FDA) show fluorescence under a GFP filter if viable, while non-viable ones appear dark. Their viability was assessed at 1 and 48 h post-isolation (*n* = 2). Transfection efficiency at 24 h was evidenced by *Green Fluorescent Protein (GFP)* expression from the reporter gene contained in the vector. (**B**) A graphical representation of the *OsDST* gene depicting the target zone that was selected for double-cut editing using two guide RNAs. (**C**) PCR amplifications of the *OsDST* gene fragment, including the editing target zone using genomic DNA extracted from Ónix protoplasts 24 h post-transfection without the vector (DST-wt) and with the pHUE411-DST double-cut editing vector (DST-ed). (**D**) Results of the sequencing of the *OsDST* wild-type and edited amplicons that were detected in panel **C**. Scale bars in **A**: white = 200 µm; gray = 50 µm. The lane details in **C** are as follows: ladder 1 kb Plus (ThermoFisher Scientific, Waltham, MA, USA) (L); negative control H_2_O (neg); target gene for PCR control for *β-actin* from non-transfected (Ac-wt) and transfected (Ac-ed) protoplasts with an expected amplification size of 1192 bp; gene editing screening of the *OsDST* gene in non-transfected (DST-wt) and edited protoplasts (DST-ed). The expected size of the amplicon generated from the WT allele is 555 bp, while the edited amplicon is 261 bps.

**Figure 3 plants-14-02059-f003:**
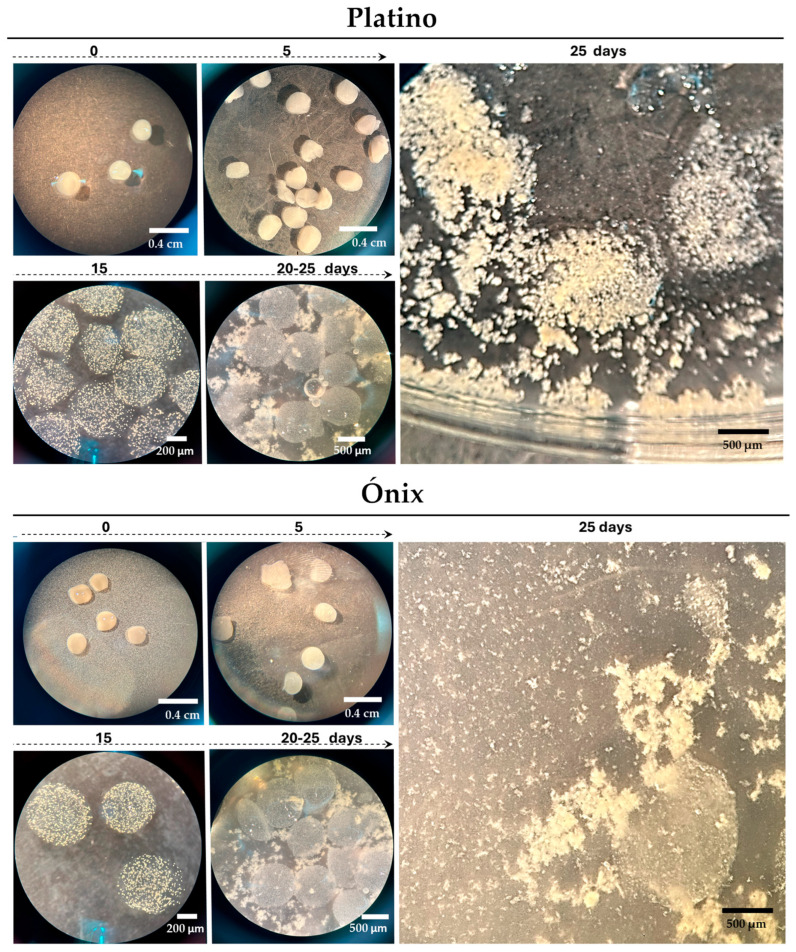
Immobilization and development of rice protoplasts in alginate beads. Proliferated protoplasts embedded in alginate beads, leading to microcalli development. The culture used was an AA medium with feeder extracts and 0.6 M mannitol for the first 15 d; we then removed the feeder and reduced mannitol to 0.3 M (15–30 d) and 0.1 M (30–45 d). The bars indicate the approximate sizes specified.

**Figure 4 plants-14-02059-f004:**
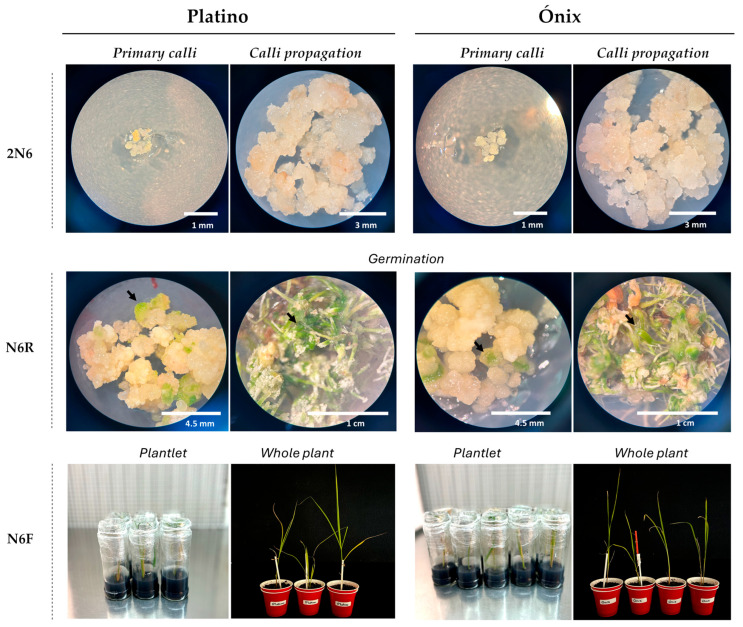
Regeneration of plants from protoplasts derived from embryogenic callus in the *japonica* rice varieties Platino and Ónix. Beginning from the transfer of the first calli, obtained from the AA liquid culture medium, to the 2N6 propagation medium, their proliferation was observed. The first germination zones (an example is indicated by an arrow, **left**) and the first sprouting zones (indicated by an arrow, **right**) are visualized in the N6R medium. Seedlings generated from the shoots are represented, followed by whole plants that are acclimatized to ex vitro conditions. The bars indicate the approximate sizes specified.

**Table 1 plants-14-02059-t001:** Culture media used for temperate *japonica* rice protoplast isolation and regeneration.

Medium Name/Purpose	Medium Composition
AA * ^1^/for induction of protoplast division and microcallus formation	CaCl_2_·2H_2_O 0.44 g/L, KH_2_PO_4_ 0.17 g/L, MgSO_4_·4H_2_O 0.37 g/L, KCI 2.94 g/L, KI 0.83 mg/L, CoCl_2_·6H_2_O 0.025 g/L, H_3_BO_3_ 6.2 mg/L, Na_2_MoO_4_·2H_2_O 0.25 mg/L, MnSO_4_·4 H_2_O 22.3 mg/L, CuSO_4_·5H_2_O 0.025 mg/L, ZnSO_4_·7H_2_O 8.6 mg/L, FeSO_4_·7H_2_O 27.85 mg/L, Na-EDTA 37.25 mg/L, inositol, 100 mg/L, nicotinic acid 0.5 mg/L, pyridoxine-HCl 0.1 mg/L, thiamine-HCl 0.5 mg/L, glycine 75 mg/L, L- glutamine 877 mg/L, L-aspartic acid 266 mg/L, L-arginine 228 mg/L, 2.4-D 2 mg/L, Kinetin 0.2 mg/L, and gibberellic acid 3 0.1 mg/L; pH 5.6
Base Medium * ^2^/foundation for 2N6, N6R, and N6F media	(NH_4_)_2_SO_4_ 463 mg/L, KNO_3_ 2.83 g/L, KH_2_PO_4_ 400 mg/L, MgSO_4_·7H_2_O 185 mg/L, CaCl_2_·2H_2_O 166 mg/L, H_3_BO_3_ 1.6 mg/L, KI 0.83 mg/L, MnSO_4_·4H_2_O 4.4 mg/L, ZnSO_4_·7H_2_O 1.5 mg/L, Na_2_EDTA 37.3 mg/L, FeSO_4_·7H_2_O 27.8 mg/L, glycine 2.0 mg/L, thiamine-HCl 1.0 mg/L, pyridoxine-HCl 0.5 mg/L, nicotinic acid 0.5 mg/L, and agar 7 g/L
2N6 * ^2^/for callus induction and propagation	Base Medium supplemented with myo-inositol 100 mg/L, L-proline 500 mg/L, 2,4-D 2.0 mg/L, casein hydrolysate 500 mg/L, and sucrose 30 g/L; pH 5.8
N6R * ^2^/for induction of shoot regeneration	Base medium supplemented with myo-inositol 100 mg/L, L-proline 500 mg/L, kinetin 0.5 mg/L, casein hydrolysate 1.0 mg/L, sorbitol 30 g/L, sucrose 20 g/L, and activated charcoal 500 mg/L; pH 5.8
N6F * ^2^/for induction of roots	Base medium supplemented with casein hydrolysate 1.0 mg/L, sorbitol 30 g/L, sucrose 15 g/L, and activated charcoal 500 mg/L; pH 5.8

* Formulations were adapted from ^1^ [36] and ^2^ [37], with minor modifications.

## Data Availability

The data featured in this study are publicly accessible within this work and the Appendix A.

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
