# Peer review of "Protoplast-Based Regeneration Enables CRISPR/Cas9 Application in Two Temperate Japonica Rice Cultivars"

_plants, 2025, doi:10.3390/plants14132059_

Round 1

Reviewer 1 Report (Previous Reviewer 2)

Comments and Suggestions for Authors

The manuscript has been significantly improved since the last version, and the authors have addressed all of my concerns. Therefore, I support the publication of this manuscript in current form.

Author Response

Response to Reviewers
Manuscript ID: plants-3670026
Title: Protoplast-Based Regeneration Enables CRISPR/Cas9 Application in Two Temperate Japonica Rice Cultivars
Authors: Barrera et al.

We extend our sincere thanks to the reviewers and editorial board for their thoughtful and constructive evaluations of our manuscript. Their insights substantially enhanced the clarity, reproducibility, and overall scientific quality of this work.

Below, we provide a detailed point-by-point response to each reviewer’s comments. All corresponding changes have been incorporated into the revised manuscript and highlighted accordingly. We trust that the updated version meets the journal’s expectations for publication.

Reviewer #1

Comment: The reviewer finds the manuscript acceptable without suggestions for changes.
Response: We thank Reviewer #1 for their supportive assessment and positive appraisal of our work.

Reviewer 2 Report (New Reviewer)

Comments and Suggestions for Authors

Please see attached

Author Response

Response to Reviewers
Manuscript ID: plants-3670026
Title: Protoplast-Based Regeneration Enables CRISPR/Cas9 Application in Two Temperate Japonica Rice Cultivars
Authors: Barrera et al.

We extend our sincere thanks to the reviewers and editorial board for their thoughtful and constructive evaluations of our manuscript. Their insights substantially enhanced the clarity, reproducibility, and overall scientific quality of this work.

Below, we provide a detailed point-by-point response to each reviewer’s comments. All corresponding changes have been incorporated into the revised manuscript and highlighted accordingly. We trust that the updated version meets the journal’s expectations for publication.

Reviewer #2

  1. Background on cultivars and rationale for using temperate japonica lines:
    Response: We expanded the Introduction (Lines 113–129) to include breeding background and agronomic relevance of Platino and Ónix. We also clarified the strategic importance of temperate japonica cultivars for Chilean agriculture and their suitability for in vitro culture and protoplast technology development.
  2. Improve flow in the Introduction:
    Response: The inclusion of the genotype background has reinforced the transitions between background, rationale, and objectives.

3–5. Figures: add scale bars, improve Y-axis labeling, zoom-in on Figure 3:
Response: Figures 1, 3, and 4 now include embedded scale bars. Figure 2’s Y-axis labels have been reformatted for clarity. Figure 3 includes a new zoom-in panel highlighting callus development final stages.

  1. Minor corrections (sentence, italics, sample size, PEG weight):
    Response: All requested changes have been implemented:
  • Incomplete sentence (Line 164) corrected.
  • Ex vitro (Line 275) and Oryza sativa (Line 475) italicized.
  • Sample size (n = 10) specified (Lines 464).
  • PEG molecular weight clarified as PEG-4000 (Line 487).

Reviewer 3 Report (New Reviewer)

Comments and Suggestions for Authors

This manuscript presents a valuable, well-developed protocol for protoplast isolation and regeneration in two temperate japonica rice cultivars, addressing a critical gap for applying New Genomic Techniques like CRISPR-Cas9 to these commercially important varieties. The methodology is robust and clearly detailed, utilizing somatic embryo-derived embryogenic callus as a reliable explant source. The study is well-supported by quantitative data (yields, viability, transfection efficiency, editing confirmation) and clear figures. The protocol's adaptability to other temperate japonica varieties is a significant advantage for breeding programs. Overall, this work provides a much-needed, reproducible, and effective platform enabling advanced biotechnological improvement of temperate japonica rice.

Author Response

Response to Reviewers
Manuscript ID: plants-3670026
Title: Protoplast-Based Regeneration Enables CRISPR/Cas9 Application in Two Temperate Japonica Rice Cultivars
Authors: Barrera et al.

We extend our sincere thanks to the reviewers and editorial board for their thoughtful and constructive evaluations of our manuscript. Their insights substantially enhanced the clarity, reproducibility, and overall scientific quality of this work.

Below, we provide a detailed point-by-point response to each reviewer’s comments. All corresponding changes have been incorporated into the revised manuscript and highlighted accordingly. We trust that the updated version meets the journal’s expectations for publication.

Reviewer #3

Comment: The reviewer commended the manuscript’s scientific merit and potential without suggesting changes.
Response: We gratefully acknowledge Reviewer #3’s positive feedback and encouragement. No additional changes were required.

Reviewer 4 Report (New Reviewer)

Comments and Suggestions for Authors

This work presents a valuable protoplast-based regeneration platform for two under-studied temperate japonica cultivars. The use of feeder extracts and alginate beads is particularly innovative. To sharpen your work's impact, you should more clearly differentiate your results from those already established in Taipei 309 or Nipponbare systems. Explicitly compare your yields, viability percentages, and editing efficiencies against these benchmarks.

To significantly enhance reproducibility, please report key reagent lot numbers, buffer pH values, and precise light-intensity settings. Furthermore, your figures should include sample sizes (n-values) and error bars for all viability, transfection, and regeneration metrics, instead of relegating these crucial details to the main text.

To strengthen claims of broad applicability, consider testing a third, genetically divergent japonica line. While your transient protoplast editing results are promising, demonstrating heritable edits in even a single regenerated T₀ plant would substantially bolster your study's significance.

Finally, consider adding a single "study overview" schematic that summarizes your workflow from protoplast isolation through regeneration and editing. This will provide readers with an at-a-glance visual of the entire process.

For detailed, line-by-line suggestions, please refer to the annotated PDF.

Author Response

Response to Reviewers
Manuscript ID: plants-3670026
Title: Protoplast-Based Regeneration Enables CRISPR/Cas9 Application in Two Temperate Japonica Rice Cultivars
Authors: Barrera et al.

We extend our sincere thanks to the reviewers and editorial board for their thoughtful and constructive evaluations of our manuscript. Their insights substantially enhanced the clarity, reproducibility, and overall scientific quality of this work.

Below, we provide a detailed point-by-point response to each reviewer’s comments. All corresponding changes have been incorporated into the revised manuscript and highlighted accordingly. We trust that the updated version meets the journal’s expectations for publication.

Reviewer #4

We sincerely thank Reviewer #4 for their comprehensive and insightful review. The following responses address both general and specific comments:

General Comments

Benchmark comparison with existing japonica systems:
Response: Comparisons with Nipponbare, Taipei 309, and Sasanishiki systems were added to the Discussion (Lines 302–320 and other sections when possible). Our method avoids complex steps like sucrose gradients yet achieves high yield (>10⁶ cells/g FW), 99% viability, and successful regeneration, highlighting its efficiency and adaptability for Chilean-adapted cultivars.

Improved reproducibility:
Response: We included reagent catalog and lot numbers, buffer pH, and light intensity in Section 4. Figures presenting viability, transfection, and regeneration outcomes have been improved.

Broader applicability and future scope:
Response: We agree that testing additional japonica genotypes and demonstrating heritable edits would strengthen the platform. These directions are now clearly stated as priorities in the revised Discussion. In addition, regarding broad applicability, although the use of a third, genetically divergent japonica line is not possible for the moment (it requires new SE development), we have extended this procedure to all the genotypes used in the previous study by Barrera at al (2024)( https://doi.org/10.3390/plants13030416). In addition, first edited DST materials are showing up using this platform, which also is matter of new works to be submitted.

Study overview schematic:
Response: We had already included a schematic (Figure as “graphical abstract”) summarizing the workflow from protoplast isolation to editing and regeneration. This visual tool enhances the manuscript’s clarity and accessibility.

Specific Comments

  1. Title revision:
    We have adopted the reviewer’s suggested title to better reflect the study’s contribution:
    “Protoplast-Based Regeneration Enables CRISPR/Cas9 Application in Two Temperate Japonica Rice Cultivars”
  2. Abstract clarity and enzyme specification:
    The Abstract now includes specific enzyme names and concentrations with improved flow.
  3. Figure 2D clarity:
    Image quality has been enhanced for better visualization of fluorescence and sequencing results.
  4. Figure 4 layout:
    Panels were reorganized with labeled subfigures and revised annotations.
  5. Conclusion expansion:
    The Conclusion was rewritten to emphasize the novelty, reproducibility, and practical impact of our method, especially for underutilized germplasm in Chilean breeding programs.
  6. Methodological details:
    Section 4 includes full technical parameters such as digestion times, media composition, and step-wise procedures (referenced in Table 1).
  7. Supplementary data relocation:
    Supplementary Table S2 was relocated to the main manuscript and renamed Table 1 to improve accessibility.
  8. Language clarity:
    Some stylistic improvements and language refinements were applied across all sections of the manuscript. The work was revised by the MDPI editing author´s service.

This manuscript is a resubmission of an earlier submission. The following is a list of the peer review reports and author responses from that submission.

Round 1

Reviewer 1 Report

Comments and Suggestions for Authors

If the manuscript aims to introduce a protocol, it is an excellent contribution. However, it is challenging to evaluate it as a research paper. The main reasons are as follows:

  1. The authors do not provide the process or rationale behind determining the isolation and culture conditions for the protoplasts.
  2. The manuscript mentions shoot regeneration efficiencies of 12 or 20 shoots/g callus in two cultivars, but there are no repetition data, making it difficult to assess reproducibility.
  3. There is no review of previous studies on rice protoplast culture or a discussion comparing the results of this study with past findings.
  4. Regarding the embedding of protoplasts, there is no reference to prior literature, nor is the process of optimizing the conditions described.

Reviewer 2 Report

Comments and Suggestions for Authors

In this manuscript, the authors described a method to regenerate plants from protoplasts of two cold-adapted rice varieties, Platino and Onix. The manuscript is well-written and easy to read. However, I have a few major issues:

1. Please fully describe how your new method improves on the existing methods, especially reference 8 and 12.

2. In line with point 1, your discussion on line 331-335 does not seem to be supported by your results. A side-by-side comparison between your method and sucrose gradient-based method is required in order to reach this conclusion

3. Figure 2, please show us both bright field and GFP images, and count how many % of cells are viable in at least three bio-reps, and include a graph of quantification.

4. Generally I think that the quality of images needs improvement. For example, Figure 3, it is hard for me to see what you are showing in the last two images on the right. Figure 4, it is hard for me to see what you are pointing to with the blue arrows. Please use higher quality images.

5. Please include scale bars for all your images.

Reviewer 3 Report

Comments and Suggestions for Authors

Review of the Manuscript
Title: Development of a protoplast-based regeneration method for two temperate japonica rice cultivars
Journal: Plants (MDPI)

Dear Authors,

I would like to present my review of the manuscript entitled "Development of a protoplast-based regeneration method for two temperate japonica rice cultivars," submitted for publication in the journal Plants (MDPI).

The manuscript addresses fundamental aspects of rice protoplast biotechnology, focusing on optimizing protocols for the isolation, transfection, and regeneration of protoplasts from embryogenic callus cultures. The study emphasizes its applicability to two cultivars associated with the National Rice Breeding Program, with the broader aim of generating new genotypes for breeding and expanding their application potential. In this context, the work aligns with the journal's scope and merits publication after minor revisions. Below, I provide my detailed comments and suggestions.

General Remarks:

The manuscript is well-written, and the introduction effectively frames the study's aims and relevance, making it accessible even to readers less familiar with rice genetics and biotechnology. While the manuscript does not contain factual inaccuracies, certain areas require minor corrections and clarifications, as outlined below.

Specific Comments:

1. Introduction:

The introduction is well-constructed, providing sufficient background on the challenges in developing new rice varieties and the significance of protoplast-based methods. No further modifications are needed here.

2. Results:

2.1. Protoplast isolation (lines 127–128):
Please ensure that information regarding the composition of the enzymatic mixture used for cell wall digestion is also included in the Materials and Methods section for clarity and reproducibility.

3. Materials and Methods:

3.1. Section Numbering:
The numbering of subsections must be corrected to follow a logical sequence. For example, Materials and Methods is currently labeled as Section 4, while subordinate sections such as "Somatic embryogenesis callus induction" are labeled as 2.2. Ensure that the numbering is consistent throughout the manuscript.

3.2. Plant Material and Tissue Culture Media (Section 4.1):
It is difficult to determine the precise media used based on the current description. While the composition and abbreviations are provided in the supplementary tables (referenced as item 35), it remains unclear who authored the media protocols. Please clarify this information within the main text to improve transparency.

3.3. Somatic Embryogenesis Callus Induction (Section 2.2):

  • Correct the numbering for this subsection.
  • Additionally, ensure that ALL abbreviations are explained upon their first appearance in the text. This is essential for improving readability and aiding manuscript navigation.

3.4. Protoplast Isolation from Embryogenic Callus Cultures (Section 2.3, should be 4.3):
Provide a detailed description of how the performance and service life of the progenitors were evaluated. This information appears to be missing in the Materials and Methods section (currently Section 2.4), making it difficult to understand the methodology.

4. Supplementary Figures:

Supplementary Figure S1 (Panels A and B):
The contrast in Panels A and B should be improved to enhance the clarity and visibility of the presented data.

Final Remarks:

The manuscript is scientifically sound and provides valuable insights into rice protoplast biotechnology. Implementing the suggested corrections will improve the clarity, consistency, and overall quality of the manuscript, making it suitable for publication in Plants.
